# Design, Synthesis and Biological Evaluation of Novel Triazole *N*-acylhydrazone Hybrids for Alzheimer’s Disease

**DOI:** 10.3390/molecules25143165

**Published:** 2020-07-10

**Authors:** Matheus de Freitas Silva, Ellen Tardelli Lima, Letizia Pruccoli, Newton G. Castro, Marcos Jorge R. Guimarães, Fernanda M. R. da Silva, Nathalia Fonseca Nadur, Luciana Luiz de Azevedo, Arthur Eugen Kümmerle, Isabella Alvim Guedes, Laurent Emmanuel Dardenne, Vanessa Silva Gontijo, Andrea Tarozzi, Claudio Viegas

**Affiliations:** 1Laboratory of Research in Medicinal Chemistry (PeQuiM), Federal University of Alfenas, Jovino Fernandes Sales Avenue, 2600, Alfenas 37130000, MG, Brazil; ellentflima17@gmail.com (E.T.L); vanessagontijo@yahoo.com.br (V.S.G.); 2Department for Life Quality Studies, Alma Mater Studiorum-University of Bologna, Corso d’Augusto 237, 47921 Rimini, Italy; letizia.pruccoli2@unibo.it; 3Laboratory of Molecular Pharmacology, Federal University of Rio de Janeiro, Avenida Carlos Chagas Filho, 373, Rio de Janeiro 21941590, RJ, Brazil; ngcastro@icb.ufrj.br (N.G.C.); mj.jorge93@gmail.com (M.J.R.G.); dasilvafmr@gmail.com (F.M.R.d.S.); 4Laboratory of Molecular Diversity and Medicinal Chemistry (LaDMol-QM), Federal Rural University of Rio de Janeiro—UFRRJ, BR-465, Km 7 Seropédica-Rio de Janeiro 23890000, RJ, Brazil; nathaliafn18@gmail.com (N.F.N.); lucianaluizazevedo@gmail.com (L.L.d.A.); akummerle@yahoo.com.br (A.E.K.); 5Grupo de Modelagem Molecular em Sistemas Biológicos (GMMSB), National Laboratory for Scientific Computing—LNCC, Avenida Getúlio Vargas, 333, Petrópolis 25651-076, RJ, Brazil; isabella.alvimg@gmail.com (I.A.G.); dardenne@lncc.br (L.E.D.)

**Keywords:** molecular hybridization, Alzheimer’s disease, curcumin, resveratrol

## Abstract

Alzheimer’s disease (AD) is a multifactorial neurodegenerative disorder that involves different pathogenic mechanisms. In this regard, the goal of this study was the design and synthesis of new compounds with multifunctional pharmacological activity by molecular hybridization of structural fragments of curcumin and resveratrol connected by an *N*-acyl-hydrazone function linked to a 1,4-disubstituted triazole system. Among these hybrid compounds, derivative **3e** showed the ability to inhibit acetylcholinesterase activity, the intracellular formation of reactive oxygen species as well as the neurotoxicity elicited by Aβ_42_ oligomers in neuronal SH-SY5Y cells. In parallel, compound **3e** showed a good profile of safety and ADME parameters. Taken together, these results suggest that **3e** could be considered a lead compound for the further development of AD therapeutics.

## 1. Introduction

Alzheimer’s disease (AD) is an age-related neurological disorder with complex pathophysiology and clinical symptoms that causes memory loss, decline in thinking ability, motor and executive functions. The hallmark brain changes related to AD are due to loss of inter-neuronal connectivity and neuronal death, especially in areas such as the frontal cortex and hippocampus [1]. The pathology begins with a series of events at the molecular level, which results in the production of a functional mismatch between different neurotransmitter systems, especially in the cholinergic system, causing the peculiar cognitive and psychiatric impairments of the disease [2]. At the molecular level, acetylcholine (ACh) deficits are a response to the loss of cholinergiC-Neurons and the presence of acetylcholinesterase (AChE) levels in the synaptic clefts that hydrolyze ACh into choline and acetate [3,4]. Aggregation and deposition of β-amyloid (Aβ) peptide fragments in the neuronal extracellular environment are currently accepted as an important cause of AD. Due to a not yet clearly understood dysfunction, enzymes such as β- and γ-secretases can cleave the amyloid precursor protein (APP), generating Aβ_40_ and Aβ_42_ peptides that have a high aggregation capacity, producing insoluble protofibrils, responsible for the inflammatory process, among other pathological effects. Recent studies suggest that Aβ oligomers diffuse into the brain parenchyma and alter synaptic function, with selective neuronal loss in the cortex and hippocampus, regions characteristically affected in AD [5,6,7]. Among the mechanisms of neurotoxicity mediated by Aβ peptides, the formation of reactive oxygen species (ROS) and oxidative stress (OS) contribute to neuronal death. Several studies indicate that redox metals and mitochondrial respiratory chain activity can impair the secretase activity with the formation of Aβ creating a vicious circle that reinforces both the OS and neuronal death in AD [8,9]. In addition, the cerebral inflammatory process plays a very important role in the pathology of AD, involving several inflammatory mediators, such as cytokines, chemokines and complement proteins, influencing the formation of senile plaques. The presence of senile plaques in the brain triggers the activation of microglia and then an inflammatory process through the interleukin 1, interleukin 6, tumour necrosis factor α and cyclooxygenases-1 and 2. The inflammation at brain level also promotes an increase of APP expression and formation of new senile plaques [10,11,12].

AD shows a profile of multifactorial hallmarks involving physiological, biochemical and chemical changes, mediated by different activation pathways [13]. In this regard, there are several therapeutic strategies for treating the illness condition such as the use of two or more associated drugs or a single drug-containing structural attribute that guarantee it to act in desired multiple targets. This latest one is a strategy based on the concept of polypharmacology, and in the last decade has aroused the interest of the scientific community as a new paradigm in the search for rationally designed molecules for chronic diseases like AD [14]. In this context, natural products have been used as structural models in the drug design of ligands against AD, as evidenced by the high volume of studies published in the literature [15,16,17,18,19]. 

Curcumin (**1**) and resveratrol (**2**, Figure 1) are prominent examples of natural products with potential action against AD, given the multiple activity profiles of themselves and their derivatives reported in the literature, acting on Aβ, OS, AChE inhibition and neuroprotection [20,21]. Herein we report the synthesis and pharmacological evaluation of a family of compounds **3a**–**3m** (Figure 1), rationally designed by molecular hybridization (MH) of the structures of curcumin and resveratrol, to obtain novel molecules with multifunctional antioxidant, anti-inflammatory, neuroprotective and AChE inhibitory properties. The hybrid series **3a**–**3m** was planned by the combination of the ring patterns **A**, **B** and **C** from prototypes **1** and **2** as initial inspiration, adding different substituents in rings **B** and **C** in order to evaluate potential structure-activity relationships. As a spacer subunit, an *N*-acylhydrazone subunit coupled to a 1,4-disubstituted 1,2,3-triazole ring was used, with the expectancy that this structural pattern guarantees a relative conformational freedom by the presence of an *sp3* carbon atom between ring **A** and the triazole subunit, as well as the insertion of additional H-bond donor and acceptors sites.

## 2. Results and Discussion

### 2.1. Chemistry

The synthesis of triazole *N*-acylhydrazone derivatives **3a**–**m** comprised five steps: Initially, vanillic aldehyde (**4**) was reduced to the corresponding primary alcohol **5** by reaction with sodium borohydride in the presence of activated carbon in THF and water [22]. Alcohol **5** was reacted with thionyl chloride in dichloromethane leading to the formation of the correspondent reactive chloride, which is then converted to the azide **6 [23]**. A click reaction between intermediate azide **6** and ethyl propiolate (**7**) lead to the formation of the correspondent triazole ester **8**, which in turn was converted to the intermediate hydrazide **9** by reaction with hydrazine monohydrate [24,25]. Finally the hydrazide **9** was coupled with different functionalized aldehydes **10** to give the desired triazole *N*-acylhydrazone derivatives **3a**–**m** (Figure 2) [12].

### 2.2. Biological Evaluation and Computational Study

#### 2.2.1. Evaluation of AChE Inhibition

Initially, all compounds were screened at a standard screening concentration of 30 μM and the resulting percentages of AChE inhibition by the triazole derivatives **3a**–**m** are shown in Figure 3. Compound **3e** was identified as the most active in the series, inhibiting more than 50% of the AChE activity at the screening concentration and exhibiting an IC_50_ value of 26.30 μM, being more potent than curcumin (IC_50_= 132.13 μM [26]), but less potent than donepezil (IC_50=_ 0.026 μM [26]). A study was also performed to verify the ability of **3e** to inhibit BuChE, but no relevant activity was detected, being selective against AChE.

For a better understanding of the results towards AChE inhibition by the series of hybrid compounds, a docking study was performed with the most active compounds **3b** and **3e**, and for comparison purposes, with the weaker AChE inhibitor **3d**. The docking score results are shown in Table 1.

According to the ensemble docking results, the targeted compounds bind to the AChE cavity with the 4-hydroxy-3-methoxyphenyl ring from the intermediate hydrazide positioned at the bottom of the catalytic gorge, packed between the aromatic residues Trp84 and Phe330 through π-stacking interactions and the hydroxyl group donating a hydrogen bond to the His440 main chain (~3.0 Å). On the other side, the substituted phenyl ring from the hydrazone subunit is more exposed to the solvent near the peripheral anionic site (PAS) at the entrance of the gorge.

The interaction of compound **3b** was predicted with similar docking scores against the AChE conformations *4EY7* and *1ZGC* (−10.69 kcal/mol and −10.67 kcal/mol, respectively), but with distinct binding modes. In the binding mode of **3b** against the 4EY7 conformation, the 2-methoxyphenol subunit is not capable of interacting with the Trp84 side chain at the bottom of the binding site, while the resorcinol moiety is exposed to the solvent (Figure 4A). We will focus the discussion for the docked pose of **3b** against the 1ZGC conformation, which has the 4-hydroxy-3-methoxyphenyl ring from the intermediate hydrazide interacting at the bottom of the enzymatic cavity similarly to the donepezil binding mode (Figure 4B). In the docking pose of compound **3b** with 1ZGC (docking score = −10.67 kcal/mol, Figure 4B), one hydroxyl group of the resorcinol moiety attached at the *meta* position makes two hydrogen bond interactions with AChE, donating one hydrogen bond to the Ser286 side chain and accepting another one from the Arg289 main chain, whereas the other hydroxyl group is exposed to the solvent. The compound **3e**, which contains only one hydroxyl group at the para position, is capable of making only one hydrogen bond with the Ser286 side chain (2.97 Å) whereas the methoxy group is exposed to the solvent. (Figure 4C).

For compound **3d**, the presence of two methoxy groups attached to the phenyl ring exposed to the solvent and the absence of hydrogen bonds with PAS (Figure 4D), might be deleterious for its binding affinity against AChE when compared to **3b** and **3e**.Comparative analysis with potent AChE inhibitors described in the literature, such as donepezil, also indicates that the absence of a positively charged group, important to create a dipole moment compatible to that observed for the AChE gorge [27,28] and to make cation-π interactions with aromatic residues at the bottom of the cavity, together with the high desolvation cost of polar groups not performing hydrogen bonds with the binding site present in these compounds (i.e., NAH and triazole group), could be related with the weaker binding affinities predicted for this class of compounds.

#### 2.2.2. Toxicity

Initially, we evaluated the toxic effects of compounds **3a**–**m** at different concentrations (2.5-80 μM) in human neuronal (SH-SY5Y) cells after 24 h of treatment by 3-(4,5-dimethylthiazol-2-yl)-2,5-diphenyltetrazolium bromide (MTT) assay [29]. As shown in Figure 5, some compounds, such as **3c** (IC_50_ = 10.30 μM), **3g** (IC_50_ = 10.35 μM), **3h** (IC_50_ = 3.02 μM), **3i** (IC_50_ =10.40 μM) and **3k** (IC_50_ = 3.62 μM), exerted a strong toxicity, while compounds, **3a**, **3b** and **3e**, did not show any toxic effects at 80 μM. Among the compounds not associated to toxicity, and considering its best result on AChE inhibition, we selected **3e** to also evaluate its cytotoxicity in non-neuronal cells.

We therefore selected the compounds **3a**, **3b** and **3e** without toxicity to perform the following assays in SH-SY5Y cells.

#### 2.2.3. In Silico ADME Properties

In parallel to evaluation of hybrid compound neurotoxicity we performed an in silico prediction study of ADME (Table 2) properties with the QikProp tool from Maestro (Schrödinger Release 2018-4, Schrödinger, LLC, New York, NY, USA, 2018). In particular, the hybrid compounds **3a**, **3b** and **3e** that did not show any toxic effect at the neuronal level also showed acceptable properties predicted of oral absorption as well as permeation of the blood-brain barrier. In addition, the Lipinski’s rule of five suggested the potential druggability of **3a**, **3b** and **3e**.

#### 2.2.4. Antioxidant Activity

As reported in the Introduction, the OS plays a critical role in several neurodegenerative processes of AD. In this context, we evaluated the direct and indirect antioxidant activity of the compounds in terms of ability to scavenge the radical species and increase the intracellular antioxidant defenses, respectively, in SH-SY5Y cells.

Initially, the direct antioxidant activities of all compounds **3a**–**m** (1.56–200 mg/L) were assessed against the DPPH radical in the absence of neuronal cells. Ascorbic acid and Trolox were used as reference compounds (Table 3).

The compounds **3a**–**d** and **3f**–**m** did not show relevant direct antioxidant activity. In contrast the compound **3e** scavenged the DPPH radical in a similar manner to the Trolox positive control (**3e**, EC_50_ = 30.44 μM vs Trolox, EC_50_ = 27.76 μM). In parallel, we evaluated the antioxidant activity of the compounds not associated with toxicity (Figure 5) against the ROS formation induced by *t*-BOOH in SH-SY5Y cells. To appraise the direct and indirect antioxidant activity, the SH-SY5Y cells were simultaneously treated for 30 min with **3a**, **3b** and **3e** (10 µM) and *t*-BOOH (100 µM) or chronically treated for 24 h (the time necessary to activate the endogenous antioxidant system) with **3a**, **3b** and **3e** (10 µM) and then exposed to *t*-BOOH (100 µM) for 30 min, respectively. Then, the intracellular ROS formation was detected by using the fluorescent probe 2′,7′-dichlorodihydrofluorescein diacetate (H_2_DCF-DA). As shown in Table 2, compound **3e**, but not **3a** and **3b**, exerted direct antioxidant effects against the ROS formation evoked by *t*-BOOH in SH-SY5Y cells, confirming the data obtained with DPPH assay. All tested compounds showed indirect antioxidant effects, in terms of increased neuronal resistance against the ROS formation, with a maximum activity for **3e** (35.16%). Taken together, these results suggest that compound **3e** exhibits both direct and indirect antioxidant activity against the oxidative stress in neuronal cells.

Among the endogenous antioxidants in the central nervous system, glutathione (GSH) plays an important role to maintain the redox balance by directly interacting with ROS formed during the physiological metabolic processes [30,31,32]. Recent studies suggest that the OS increase in AD is also associated with low levels of endogenous antioxidants, including GSH, in the brain. We therefore evaluated the ability of compounds **3a**, **3b** and **3e** to modulate the intracellular GSH levels in SH-SY5Y cells, after 24 h of treatment using the fluorescent probe monochlorobimane (MCB). The treatment of SH-SY5Y cells for 24 h with the compounds increased the GSH levels, with a maximum of percentage increase for **3e** (Figure 6). These results confirm the profile of indirect antioxidant activity described in Table 3.

In this regard, **3a**, **3b** and **3e** have structural differences that justify their different antioxidant activity. Among the hybrid compounds, **3e** shows the best aspect of free radical resonance: (i) **3a** and **3e** can relocate the radical from the phenolic hydroxyl to the *N* atom of the N-acylhydrazone group, due to the hydroxyl’s para position in relation to the *sp^2^* carbon of N-acylhydrazone; (ii) **3a** shows less resonance hybrids than **3e**, due to the absence of methoxy next to the phenolic hydroxyl; (iii) **3b** does not have the ability to relocate the free radical outside the aromatic ring or between the hydroxyls on the same aromatic ring.

#### 2.2.5. Neuroprotective Activity

Given the best antioxidant profile of compound **3e** we also evaluated its ability to protect the SH-SY5Y cells against the neurotoxicity elicited by Aβ_42_ peptide oligomers (OAβ_42_), soluble aggregates of Aβ peptides involved in the pathogenesis of AD. As shown in Figure 7, the concomitant treatment of SH-SY5Y cells with **3e** (5 µM) for 4 h significantly reduced the loss of neuronal viability induced by OAβ_42_ (10 µM). 

The results suggest that hybrid compound **3e** could halt the interaction between the OAβ_1–42_ and the neuronal membrane, an early step of the neurotoxicity mediated by OAβ_42_.

## 3. Conclusions

Among these new triazole *N*-acylhydrazone hybrid series, compound **3e** showed a potential profile of a multifunctional compound with the ability to inhibit AChE activity and counteract the ROS formation through direct and indirect antioxidant mechanisms. In particular, some structural attributes of **3e**, such as the position of the hydroxyls on aromatic ring, were indispensable for its antioxidant activity in neuronal cells. This compound also showed a good profile of safety in the same neuronal model and in silico ADME parameters. Taken together, these results suggest that compound **3e** could be considered a lead compound for the development of further AD therapeutics.

## 4. Experimental Section

### 4.1. General Information

IR spectra were generated on a Nicolet iSso infrared spectrometer (Thermo Scientific, Madison, WI, USA) coupled to Gladi ATR device (Pike, Madison, WI, USA) at the Analysis and Characterization of Drugs Laboratory (LACFar) at the Federal University of Alfenas (UNIFAL-MG). The ^1^H- and ^13^C-NMR spectra were obtained on an AC-300 spectrometer (Bruker, Billerica, MA, USA) operating at 300 MHz for ^1^H-NMR and 75 MHz for ^13^C-NMR at the Nuclear Magnetic Resonance Laboratory at the UNIFAL-MG. Chromatograms for purity testing of the final compounds were recorded on a Prominence model LC-10 HPLC (Shimadzu, Tokyo, Japan) equipped with an automatic injector, a 300 nm UV/VIS detector and a C_18_ analytical column (Shimadzu CLC-ODS, 250 mm × 4.4 mm, inner diameter × 5 µm). The analyses were conducted in a two-phase isocratic system with a constant concentration of 90% methanol (A) and 10% acetonitrile (B). Assays were conducted at 20 °C at a flow rate of 0.7 mL/min. All reagents used in synthesis were purchased from Sigma-Aldrich (St. Louis, MO, USA) and used without further purifications. Thin-layer chromatography experiments were performed on silica gel sheet 60 F_254_ (Merck, Darmstadt, Germany) and chromatography column was performed on flash silica gel (220–440 mesh, 0.035–0.075 mm, Sigma-Aldrich). The visualization of the substances was done in an UV chamber (λ = 254 or 365 nm). The solvents dichloromethane, ethanol, and dimethylformamide were distilled and dried according to literature [33]. Melting points were determined on Mars equipment (Pulse Frequency Modulation, PFM II) with crushed samples packed into capillary tubes and are reported without correction. All spectra, including those of intermediates, are available in the Appendix A.

### 4.2. Chemistry

#### 4.2.1. 4-(Hydroxymethyl)-2-methoxyphenol (**5**)

Initially, a solution of vanillic aldehyde (2 g, 13.15 mmol, 1 eq.) in THF (10 mL) was placed in a round bottomed flask under vigorous stirring, followed by the addition of activated carbon (158 mg, 1 eq.) and NaBH_4_ (249 mg, 0.5 eq.). The resulting solution was stirred for 1 min and then 2 mL of distilled H_2_O were slowly added to the reaction. The reaction mixture was kept under stirring for a further 2 min. Finally, an additional 5 mL of distilled H_2_O was added and the solution was stirred for a further 5 min until total consumption of starting material [22]. For isolation of the desired alcohol, 10 mL of distilled H_2_O was added and the aqueous solution was filtered off the activated carbon and exhaustively extracted with CH_2_Cl_2_. The organic phase was concentrated and the pure product was obtained as white crystals (1.763 g, 11.44 mmol).

#### 4.2.2. 4-(Azidomethyl)-2-methoxyphenol (**6**) 

In a round bottomed flask, alcohol **5** (1 g, 6.50 mmol, 1 eq.) was reacted with thionyl chloride (1 mL, 2 eq.) and DMF (4 drops) in CH_2_Cl_2_ (15 mL). The reaction mixture was kept under magnetic stirring for approximately two hours until TLC analysis indicated the completion of the reaction. The solvent and the excess of thionyl chloride were removed under vacuum. Subsequently, a 0.5 M solution of NaN_3_ (26 mL, 13 mmol, 2 eq.) in DMSO was added to the flask, and the resulting solution was kept under magnetic stirring for approximately 16 h [23]. When TLC analysis indicated the end of the reaction, distilled H_2_O (20 mL) was added, followed by exhaustive extraction with chloroform. The organic phase was concentrated under vacuum and the resulting crude product was purified by flash chromatography in silica gel, eluted with ethyl acetate: hexane (9:1) to give a brown oil (827 mg, 4.62 mmol).

#### 4.2.3. Ethyl 1-(4-hydroxy-3-methoxybenzyl)-1H-1,2,3-triazole-4-carboxylate (**8**) 

In a round bottomed flask, azide **6** (375 mg, 2.09 mmol, 1 eq.) was dissolved in a mixture of H_2_O/CH_2_Cl_2_ (8 mL, 1:1 *v*:*v*), followed by the addition of ethyl propiolate (7.205 mg, 1 eq.), CuSO_4_ (133 mg, 0.4 eq.) and sodium ascorbate (83 mg, 0.2 eq.). The reaction mixture was kept under magnetic stirring at 20 °C, for 20 h [24]. The reaction product was isolated by adding distilled H_2_O (5 mL), followed by extraction with CH_2_Cl_2_ (3 × 8 mL). The resulting organic phase was washed with an aqueous solution of EDTA 50% *v*/*v* and concentrated NH_4_OH (1:1) for removing the residual Cu(II). The combined organic phases were dried with anhydrous Na_2_SO_4_ and, after filtration, the solvent was removed under reduced pressure. The crude product was purified by column chromatography in silica gel, eluted with 100% CH_2_Cl_2_, affording the correspondent triazole **8** (365 mg, 1.32 mmol) as a yellowish solid.

#### 4.2.4. 1-(4-Hydroxy-3-methoxybenzyl)-1H-1,2,3-triazole-4-carbohydrazide (**9**) 

In a round bottomed flask, the triazole **8** (300 mg, 1.08 mmol, 1 eq.) was dissolved in ethanol (10 mL), followed by the addition of hydrazine monohydrate (0.640 mL, 10 eq.). The reaction mixture was kept under magnetic stirring at 20 °C for 17 h [25]. After TLC analysis indicated the total consumption of the starting material, the reaction flask was refrigerated for 5 h and the reaction product was isolated by filtration, washed with cold ethanol, providing the desired hydrazide **9** (200 mg, 0.756 mmol) as a pale yellow solid.

#### 4.2.5. General procedure for obtaining the final triazoles 3**a**–**m**


In a round bottom flask, the appropriate aldehyde **10** (1.2 eq.) was dissolved in dry ethanol (10 mL), followed by the addition of concentrated HCl (5 eq.). Then, hydrazide **9** (100 mg, 0.3801 mmol, 1 eq.), previously solubilized in the smallest possible volume of dry ethanol, was added. The resulting solution was kept under magnetic stirring and 20 °C for 24h [12]. After TLC analysis indicated the end of the reaction, the pure triazole derivative was obtained after filtration and exhaustive washing with ice-cold ethanol.

*(E)-1-(4-Hydroxy-3-methoxybenzyl)-N’-(4-hydroxybenzylidene)-1H-1,2,3-triazole-4-carbohydrazide* (**3a**): Yellow solid (yield 20%), m.p. 274 °C, purity: 99.1%. IR (ATR): *ν* 3331, 3237, 1645, 1600, 1433, 1367 and 1036 cm^−1^. ^1^H-NMR (DMSO-*d_6_*) *δ* 11.83 (*s*, 1H, N-NH); 9.93 (*s*, 1H, OH), 9.13 (*s*, 1H, OH), 8.72 (*s*, 1H, N-CH=C), 8.41 (*s*, 1H, N=CH), 7.53 (*d*, *J* = 8.61 Hz, 2H, Ar-H), 7.05 (*d*, *J* = 1.59 Hz, 1H, Ar-H), 6.87–6.75 (*m*, 4H, Ar-H), 5.53 (*s*, 2H, Ar-CH_2_-N) and 3.77 (*s*, 3H, O-CH_3_). ^13^C-NMR (DMSO-*d_6_*) *δ* 159.3, 155.8, 148.4, 147.6, 146.7, 142.0, 128.8, 126.9, 126.0, 125.2, 121.1, 115.6, 115.5, 112.6, 55.5 and 53.2. HR-MS (ESI) *m/z*: Calcd for [M + H]^+^ 368.1359 g/mol, found 368.1351 g/mol.

*(E)-N’-(3,5-Dihydroxybenzylidene)-1-(4-hydroxy-3-methoxybenzyl)-1H-1,2,3-triazole-4-carbohydrazide* (**3b**): Brown solid (yield 20%), m.p. 185 °C, purity: 98.1%. IR (ATR): *ν* 3536, 3217, 1671, 1590, 1470, 1323 and 1009 cm^−1^. ^1^H-NMR (DMSO-*d_6_*) *δ* 11.97 (*s*, 1H, N-NH), 9.50 (*s*, 2H, OH), 9.20 (*s*, 1H, OH), 8.77 (*s*, 1H, N-CH=C), 8.33 (*s*, 1H, N=CH), 7.05 (*s*, 1H, Ar-H), 6.87–6.73 (*m*, 2H, Ar-H), 6.58 (*s*, 2H, Ar-H), 6.28 (*s*, 1H, Ar-H), 5.54 (*s*, 2H, Ar-CH_2_-N), 3.77 (*s*, 3H, O-CH_3_). ^13^C-NMR (DMSO-*d_6_*) *δ* 158.6, 156.0, 148.5, 147.6, 146.7, 141.8, 135.9, 127.0, 126.0, 121.1, 115.5, 112.6, 105.1, 104.4, 55.5 and 53.2. HR-MS (ESI) *m/z*: Calcd for [M + H]^+^ 384.1308 g/mol, found 384.1292 g/mol.

*(E)-1-(4-Hydroxy-3-methoxybenzyl)-N’-(4-methoxybenzylidene)-1H-1,2,3-triazole-4-carbohydrazide* (**3c**): Light yellow solid (yield 22%), m.p. 196 °C, purity: 95%. IR (ATR): *ν* 3340, 3298, 1652, 1600, 1436, 1370 and 1022. cm^−1^. ^1^H-NMR (DMSO-*d_6_*) *δ* 11.91 (*s*, 1H, N-NH); 8.74 (*s*, 1H, N-CH=C); 8.64 (*s*, 1H, OH), 8.47 (*s*, 1H, N=CH); 7.82 (*d*, *J* = 8.81 Hz, 2H, Ar-H); 7.64 (*d*, *J* = 8.77 Hz, 2H, Ar-H); 7.10–6.97 (*m*, 3H, Ar-H); 5.54 (*s*, 2H, Ar-CH_2_-N); 3.81 (*s*, 3H, O-CH_3_) and 3.77 (*s*, 3H, O-CH_3_). ^13^C-NMR (DMSO-*d_6_*) *δ* 161.6; 160.4; 155.9; 148.0; 146.7; 141.9; 129.9; 128.6; 127.0; 126.0; 121.1; 115.5; 114.2; 112.6; 55.5; 55.2 and 53.2. HR-MS (ESI) *m/z*: Calcd for [M + H]^+^ 382.1515 g/mol, found 382.1494 g/mol.

*(E)-N’-(3,5-Dimethoxybenzylidene)-1-(4-hydroxy-3-methoxybenzyl)-1H-1,2,3-triazole-4-carbohydrazide* (**3d**): Light yellow solid (yield 20%), m.p. 220 °C, purity: 97.9%. IR (ATR): *ν* 3413, 3243, 1647, 1591, 1453, 1310 and 1034 cm^−1^. ^1^H-NMR (DMSO-*d_6_*) *δ* 12.06 (*s*, 1H, N-NH), 9.15 (*s*, 1H, OH), 8.76 (*s*, 1H, N-CH=C), 8,64 and 8,45 (*s*, 1H, N=CH), 7.06 (*d*, *J* = 2,25 Hz, 2H, Ar-H), 6.86–6.78 (*m*, 3H, Ar-H), 6.67 (*t*, *J* = 2.28 Hz, 1H, Ar-H), 5.54 (*s*, 2H, Ar-CH_2_-N), 3.80 (*s*, 3H, O-CH_3_), 3.79 (*s*, 6H, O-CH_3_) and 3.77 (*s*, 3H, O-CH_3_). ^13^C-NMR (DMSO-*d_6_*) *δ* 161.5, 160.6, 156.1, 148.0, 146.7, 141.7, 136.2, 127.1, 121.1, 115.5, 112.6, 106.0, 104.7, 102.3, 55.3 and 53.2. HR-MS (ESI) *m/z*: Calcd for [M + H]^+^ 412.1621 g/mol, found 412.1603 g/mol.

*(E)-1-(4-Hydroxy-3-methoxybenzyl)-N’-(4-hydroxy-3-methoxybenzylidene)-1H-1,2,3-triazole-4-carbo-hydrazide* (**3e**): Light yellow solid (yield 22%), m.p. 222 °C, purity: 99.9%. IR (ATR): *ν* 3473, 3346, 3255, 1652, 1601, 1572, 1465, 1278 and 1027 cm^−1^. ^1^H-NMR (300 MHz, DMSO-*d_6_*) *δ* 11.90 (*s*, 1H, N-NH), 9.62 (*s*, 1H, OH), 9.22 (*s*, 1H, OH), 8.74 (*s*, 1H, N-CH=C), 8.39 (N=CH), 7.29 (*d*, *J* = 1.55 Hz, 1H, Ar-H), 7.07–7.01 (*m*, 2H, Ar-H), 6.85-6.80 (*m*, 2H, Ar-H), 6.75 (*d*, *J* = 8.05 Hz, 1H, Ar-H), 5.53 (*s*, 2H, Ar-CH_2_-N), 3.82 (*s*, 3H, CH_3_) and 3.76 (*s*, 3H, CH_3_). ^13^C-NMR (DMSO-*d_6_*) *δ* 156.1, 149.1, 148.9, 148.1, 147.8, 146.9, 142.1, 127.2, 126.2, 125.8, 122.4, 121.3, 115.7, 115.5, 112.8, 109.0, 55.7, 55.6 and 53.4. HR-MS (ESI) *m/z*: Calcd for [M + H]^+^ 398.1464 g/mol, found 398.1446 g/mol.

*(E)-N’-Benzylidene-1-(4-hydroxy-3-methoxybenzyl)-1H-1,2,3-triazole-4-carbohydrazide* (**3f**) Light brown solid (yield 17%), m.p. 215°C, purity: 96.8%. IR (ATR): *ν* 3376, 3264, 1662, 1567, 1435, 1373 and 1044 cm^−1^. ^1^H-NMR (DMSO-*d_6_*) *δ* 8.78 (*s*, 1H, N-CH=C), 8.54 (*s*, 1H, N=CH), 7.70 (*d*, *J* = 5.42 Hz, 2H, Ar-H), 7.48–7.41 (*m*, 3H, Ar-H), 7.06 (*s*, 1H, Ar-H), 6.81 (*dd*, *J* = 8.15 and 18.24 Hz, 2H, Ar-H), 5.54 (*s*, 2H, Ar-CH_2_-N), and 3.77 (*s*, 3H, O-CH_3_). ^13^C-NMR (DMSO-*d_6_*) *δ* 156.1, 148.1, 147.6, 146.7, 141.8, 134.2, 130.0, 128.7, 127.1, 127.0, 126.0, 121.1, 115.5, 112.6, 55.5 and 53.2. HR-MS (ESI) *m/z*: Calcd for [M+Na]^+^ 374.1229 g/mol, found 374.1217 g/mol.

*(E)-1-(4-Hydroxy-3-methoxybenzyl)-N’-(4-(piperidin-1-yl)benzylidene)-1H-1,2,3-triazole-4-carbohydrazide* (**3g**): Yellow solid (yield 23%), m.p. 225°C, purity: 97.4%. IR (ATR): *ν* 3386, 3281, 2425, 2357, 1663, 1570, 1451, 1276 and 1011 cm^−1^. ^1^H-NMR (DMSO-*d_6_*) *δ* 12.05 and 11.68 (*s*, 1H, N-NH), 8.87 (*s*, 1H, N-CH=C), 8.78 and 8.49 (*s*, 1H, N=CH), 7.96–7.47 (*m*, 4H, Ar-H), 7.05 (*s*, 1H, Ar-H), 6.86-6.76 (*m*, 3H, Ar-H), 5.55 (*s*, 2H, Ar-CH_2_-N), 3.76 (*s*, 3H, O-CH_3_), 3.52-3.36 (*m*, 4H, Pip-H) and 1.94-1.53 (*m*, 6H, Pip-H). ^13^C-NMR (DMSO-*d_6_*) *δ* 158.9, 156.0, 147.6, 146.8, 141.8, 139.4, 128.3, 127.5, 127.0, 126.0, 125.8, 121.1, 115.5, 112.7, 55.6, 53.2, 23.7 and 22.0. HR-MS (ESI) *m/z*: Calcd for [M + H]^+^ 435.2145 g/mol, found 435.2119 g/mol.

*(E)-N’-(4-Chlorobenzylidene)-1-(4-hydroxy-3-methoxybenzyl)-1H-1,2,3-triazole-4-carbohydrazide* (**3h**): White solid (yield 22%), m.p. 268 °C, purity: 99.9%. IR (ATR): *ν* 3427, 3263, 3066, 1662, 1607, 1553, 1461, 1240 and 1029 cm^−1^. ^1^H-NMR (DMSO-*d_6_*) *δ* 12.16 (*s*, 1H, N-NH), 9.17 (*s*, 1H, OH), 8.78 (*s*, 1H, N-CH=C), 8.52 (*s*, 1H, N=CH), 7.72 (*d*, *J* = 8.32 Hz, 2H, Ar-H), 7.52 (*d*, *J* = 8.32 Hz, 2H, Ar-H), 7.06 (*s*, 1H, Ar-H), 6.91–6.69 (*m*, 2H, Ar-H), 5.54 (*s*, 2H, Ar-CH_2_-N), and 3.77 (*s*, 3H, O-CH_3_). ^13^C-NMR (DMSO-*d_6_*) *δ* 156.1, 147.6, 146.8, 146.7, 141.7, 134.4, 133.2, 128.9, 128.6, 127.2, 126.0, 121.1, 115.5, 112.6, 55.5 and 53.2. HR-MS (ESI) *m/z*: Calcd for [M + H]^+^ 386.1020 g/mol, found 368.1000 g/mol.

*(E)-N’-(3-Chlorobenzylidene)-1-(4-hydroxy-3-methoxybenzyl)-1H-1,2,3-triazole-4-carbohydrazide* (**3i**): White solid (yield 21%), m.p. 234 °C, purity: 99.9%. IR (ATR): *ν* 3317, 3212, 3055, 1682, 1596, 1568, 1462, 1341, 1240 and 1034 cm^−1^. ^1^H-NMR (DMSO-*d_6_*) *δ* 12.21 (*s*, 1H, N-NH), 9.16 (*s*, 1H, OH), 8.79(*s*, 1H, N-CH=C), 8.52 (*s*, 1H, N=CH), 7.75 (*s*,1H, Ar-H), 7.64 (*d*, *J* = 3.67 Hz, 1H, Ar-H), 7.49 (*d*, *J* = 4.54 Hz, 2H, Ar-H), 7.06 (*s*, 1H, Ar-H), 6.81 (*q*, *J* = 8.24 Hz, 2H, Ar-H), 5.55 (*s*, 2H, Ar-CH_2_-N), and 3.77 (*s*, 3H, O-CH_3_). ^13^C-NMR (75 MHz, DMSO-*d_6_*) *δ* 156.2, 147.6, 146.7, 146.4, 141.6, 133.6, 130.7, 129.6, 127.3, 126.1, 126.0, 125.8, 121.1, 115.5, 112.6, 55.6 and 53.2. HR-MS (ESI) *m/z*: Calcd for [M + H]^+^ 386.1020 g/mol, found 386.0998 g/mol.

*(E)-N’-(2-Chlorobenzylidene)-1-(4-hydroxy-3-methoxybenzyl)-1H-1,2,3-triazole-4-carbohydrazide* (**3j**) White solid (yield 22%), m.p. 229 °C, purity: 99.2%. IR (ATR): *ν* 3402, 3249, 1654, 1606, 1517, 1434, 1368, 1244 and 1047 cm^-1^. ^1^H-NMR (300 MHz, DMSO-*d_6_*) *δ* 12.37 (*s*, 1H, N-NH), 9.16 (*s*, 1H, OH), 9.96 (*s*, 1H, N-CH=C), 8.78 (*s*, 1H, N=CH), 8.04-7.97 (*m*, 1H, Ar-H), 7.55–7.50 (*m*, 1H, Ar-H), 7.49-7.41 (*m*, 2H, Ar-H), 7.06 (*s*, 1H, Ar-H), 6.89-6.72 (*m*, 2H, Ar-H), 5.55 (*s*, 2H, Ar-CH_2_-N), and 3.77 (*s*, 3H, O-CH_3_). ^13^C-NMR (DMSO-*d_6_*) *δ* 156.2, 147.6, 146.7, 144.2, 141.6, 131.4, 129.8, 127.5, 127.2, 126.8, 126.0, 121.1, 115.5, 112.6, 55.6 and 53.2. HR-MS (ESI) *m/z*: Calcd for [M + H]^+^ 386.1020 g/mol, found 386.1003 g/mol.

*(E)-N’-(4-Fluorobenzylidene)-1-(4-hydroxy-3-methoxybenzyl)-1H-1,2,3-triazole-4-carbohydrazide* (**3k**): With solid (yield 20%), m.p. 257 °C, purity: 99.9%. IR (ATR): *ν* 3423, 3263, 3069, 1663, 1603, 1440, 1373, 1240 and 1030 cm^−1^. ^1^H-NMR (DMSO-*d_6_*) *δ* 12.09 (*s*, 1H, N-NH), 9.17 (*s*, 1H, OH), 8.77 (*s*, 1H, N-CH=C), 8.53 (*s*, 1H, N=CH), 7.76 (*dd*, *J* = 5.97 and 7.98 Hz, 2H, Ar-H), 7.30 (*t*, *J* = 8.66 Hz, 2H, Ar-H), 7.06 (*s*, 1H, Ar-H), 6.89–6.71 (*m*, 2H, Ar-H), 5.54 (*s*, 2H, Ar-CH_2_-N) and 3.77 (*s*, 3H, O-CH_3_). ^13^C-NMR (DMSO-*d_6_*) *δ* 163.0 (*d*, *J* = 248.08 Hz), 156.1, 147.6, 147.0, 146.7, 141.8, 130.8 (*d*, *J* = 1.98 Hz), 129.2 (*d*, *J* = 8.61 Hz), 127.1, 126.0, 121.1, 116.0, 115.6 (*d*, *J* = 14.33 Hz), 112.6, 55.5 and 53.2. HR-MS (ESI) *m/z*: Calcd for [M+Na]^+^ 392.1134 g/mol, found 392.1124 g/mol.

*(E)-N’-(2-Fluorobenzylidene)-1-(4-hydroxy-3-methoxybenzyl)-1H-1,2,3-triazole-4-carbohydrazide* (**3l**): White solid (yield 19%), m.p. 225 °C, purity: 99.9%. IR (ATR): *ν* 3373, 3261, 3065, 1651, 1567, 1450, 1370, 1239 and 1034 cm^−1^. ^1^H-NMR (DMSO-*d_6_*) *δ* 12.27 (*s*, 1H, N-NH), 9.16 (*s*, 1H, OH), 8.79 (*s*, 2H, N-CH=C and N=CH), 7.94 (*t*, *J* = 7.46 Hz, 1H, Ar-H), 7.55-7.43 (*m*, 1H, Ar-H), 7.29 (*t*, *J* = 9.06 Hz, 2H, Ar-H), 7.06 (*s*, 1H, Ar-H), 6.86–6.74 (*m*, 2H, Ar-H), 5.55 (*s*, 2H, Ar-CH_2_-N) and 3.77 (*s*, 3H, O-CH_3_). ^13^C-NMR (DMSO-*d_6_*) *δ* 160.7 (*d*, *J* = 249.86 Hz), 156.1, 147.6, 146.7, 141.6, 140.8, 131.9 (*d*, *J* = 8.17 Hz), 127.2, 126.3, 126.0, 124.8 (*d*, J = 2.43 Hz), 121.8 (*d*, *J* = 9.97 Hz), 121.1, 115.9 (*d*, *J* = 20.82 Hz), 115.5, 112.6, 55.5 and 53.2. HR-MS (ESI) *m/z*: Calcd for [M + H]^+^ 370.1315 g/mol, found 370.1292 g/mol.

*(E)-N’-(3-Fluorobenzylidene)-1-(4-hydroxy-3-methoxybenzyl)-1H-1,2,3-triazole-4-carbohydrazide* (**3m**): With solid (yield 17%), m.p. 251 °C, purity: 99.7%. IR (ATR): *ν* 3320, 3149, 3066, 2836, 1682, 1598, 1463, 1357, 1253 and 1032 cm^−1^. ^1^H-NMR (DMSO-*d_6_*) *δ* 12.19 (*s*, 1H, N-NH), 9.16 (*s*, 1H, OH), 8.78 (*s*, 1H, N-CH=C), 8.54 (*s*, 1H, N=CH), 7.55-7.46 (*m*, 3H, Ar-H), 7.31–7.24 (*m*, 1H, Ar-H), 7.06 (*s* 1H, Ar-H), 6.86-6.75 (*m,* 2H, Ar-H), 5.55 (*s*, 2H, Ar-CH_2_-N), and 3.77 (*s*, 3H, O-CH_3_). ^13^C-NMR (DMSO-*d_6_*) *δ* 162.3 (*d*, *J* = 244.08 Hz), 156.2, 147.6, 146.7, 141.7, 136.8 (*d*, *J* = 7.61 Hz), 130.8 (*d*, *J* = 7.62 Hz), 127.2, 126.0, 123.5, 121.1, 116.8 (*d*, *J* = 21.29 Hz), 115.5, 113.0, 112.7, 55.6 and 53.2. HR-MS (ESI) *m/z*: Calcd for [M+Na]^+^ 392.1134 g/mol, found 392.1113 g/mol.

### 4.3. Pharmacological Activity

#### 4.3.1. Evaluation of AChE Inhibition

For in vitro evaluation of inhibition of EeAChE (EC3.1.1.7, type V-S, purified from Electrophorus electricus) and eqBuChE (EC 3.1.1.8, purified from equine serum, both from Sigma-Aldrich), the kinetic assay was performed according to the Ellman method [34] modified using 96-well plates as previously described [35,36]. To each well of the microplate were added 20 μL enzyme (0.5 U/mL) and 5 μL 5,5’-dithiobis-(2-nitrobenzoic acid) (10 μM) in sodium phosphate buffer (0.1 M, pH 7.4), 100 μL of the evaluated compound solutions (twice as concentrated), together with 55 μL phosphate buffer (0.1 M, pH 7.4) reaching a volume of 180 μL. After 10 min, 20 μL of the appropriate substrate (acetylthiocholine or butyrylthiocholine iodide, final concentration 0.5 M) was added and the absorbance was read on a SpectraMax 250 spectrophotometer (Molecular Devices, San Jose, CA, USA) at 412 nm for 5 min at 12-s intervals. Progression curves were acquired using the Softmax PRO 5.0 software (Molecular Devices) from which the maximum hydrolysis velocity was calculated and the analysis performed with the Prism 5 software (GraphPad Inc., La Jolla, CA, USA). The compounds evaluated were solubilized in DMSO (0.05 M) and diluted in phosphate buffer to the concentrations described immediately prior to the assay. The solvent had no detectable effect at the highest concentration used (0.2% *v*/*v*).

#### 4.3.2. Determination of DPPH Scavenging Ability

The ability of compounds to sequester DPPH free radicals was evaluated according to the method described by Gontijo and co-workers with some modifications [37]. The compounds were evaluated at concentrations of 100, 50, 25 and 12.5µM. A 4 mL aliquot of the sample was added to 1 mL of the DPPH solution (0.5 mM in ethanol). The solution was vortexed and after 30 min the absorbance was measured at 517 nm. Each solution was analyzed in triplicate and the mean values were plotted to obtain the EC_50_. Trolox and ascorbic acid were used as standards. The sequestering capacity of radicals was represented as the inhibition percentage according to the Equation (1):Inhibition percentage = [(CA – SA)/CA)] × 100(1)

CA: control absorbance; SA: sample absorbance.

#### 4.3.3. Cell Cultures

Human neuronal SH-SY5Y cells were routinely grown in Dulbecco’s modified Eagle’s Medium supplemented with 10% fetal bovine serum, 2 mM L-glutamine, 50 U/mL penicillin and 50 µg/mL streptomycin at 37 °C in a humidified incubator with 5 % CO_2_.

#### 4.3.4. Determination of Neuronal Viability

SH-SY5Y cells were seeded in a 96-well plate at 2 × 10^4^ cells/well, incubated for 24 h and then treated with the compounds for 24 h. Cell viability, in terms of mitochondrial metabolic function, was evaluated by 3-(4,5-dimethylthiazol-2-yl)-2,5-diphenyltetrazolium bromide (MTT) test, as previously described [38].

#### 4.3.5. Determination of Intracellular GSH Levels

SH-SY5Y cells were seeded in a black 96-well plate at 2 × 10^4^ cells/well, incubated for 24 h and then treated with the compounds for 24 h. GSH levels were evaluated by using monochlorobimane, as previously described [39]

#### 4.3.6. Determination of Intracellular ROS Formation

SH-SY5Y cells were seeded in a 96-well plate at 2 × 10^4^ cells/well, incubated for 24 h, and then treated with compounds for 24 h before or during the treatment of 30 min with *t*-BOOH. ROS formation was evaluated by using the fluorescent probe H_2_DCF-DA, as previously described [26].

#### 4.3.7. Determination of Neuroprotective Activity Against Aβ_42_ Oligomers

Aβ_42_ peptide (AnaSpec, Fremont, CA, USA) was first dissolved in 1,1,1,3,3,3-hexafluoroisopropanol to 1 mg/mL, sonicated, incubated at room temperature for 24 h, and lyophilised to obtain an unaggregated Aβ_42_ peptide film that was solubilised with DMSO and stored at −20 °C until use. The aggregation of Aβ_42_ peptide into oligomers was performed as previously described [40]. SH-SY5Y cells were seeded in a 96-well plate at 3 × 10^4^ cells/well, incubated for 24 h, and treated with the compounds for 24 h. The cells were then treated for 4 h with **3e** [5 μM] and OAβ_1-42_ [10 μM]. The neuroprotective activity, in terms of increase in intracellular MTT granules, was measured by MTT assay, as previously described [12].

### 4.4. In Silico Studies

We evaluated the possible binding modes of the compounds in the AChE binding site through molecular docking studies. Due to significant conformational changes observed on the peripheral anionic site (PAS) [41], we selected three representative conformations of AChE following an ensemble docking strategy similar to that adopted on previous studies [42,43,44] in addition to the structure of human AChE complexed with donepezil [45] (PDB code 4EY7). The ensemble docking strategy consists in docking the compounds into each representative conformation of the receptor aiming to consider the protein flexibility [46,47,48]. The structures selected were 1ZGC [49] (*Torpedo californica*), 2CKM [50] (*Torpedo californica*), 1Q84 [51] (*Mus musculus*) and 4EY7 [45] (*Homo sapiens*). All inhibitors from the four representative conformations of AChE interact with both CAS and PAS. Conserved waters were identified through the superposition of the structures and considered explicitly during the docking experiments (Table 4).

The receptor structures were prepared with the Protein Preparation Wizard tool from the Schrödinger Suite 2018–4 [52] and the protonation states of the amino acid residues were predicted using PROPKA with pH 7. Finally, an optimization of the hydrogen bond network of the protein-ligand complexes was performed to adjust the orientation of the hydrogen atoms, followed by an energy minimization of the hydrogen atoms. The compounds were designed and prepared with LigPrep from Maestro to set up the isomers, protonation states and tautomers with Epik [53,54] at pH 7.0 ± 0.4. We applied torsional constraints to some rotatable bonds to keep the planarity observed for some compounds during the docking experiments (Figure 8). The rotatable bond from the amide group was kept fixed to the *trans* conformation on all compounds.

The ensemble docking experiments were performed with the molecular docking program Glide [55,56] from Maestro in the XP precision mode – indicated for highly flexible ligands and to reduce false-positives. All the structures were aligned to the 1ZGC conformation using the super tool from Pymol. The grid box of the receptors was centered on the native ligand present in the 1Q84 complex (X: 98.06, Y: 53.14 and Z: 22.06). We also redocked the co-crystallized ligands into their respective AChE conformation and to the non-native structures (i.e., cross-docking) to validate the docking protocol adopted in this work. We selected the ligand pose according to the lowest XP Score among all the four AChE conformations.

#### Ensemble Docking Validation

We docked the four reference ligands into all the representative AChE conformations selected for the ensemble (i.e., redocking and cross-docking experiments) to validate the ensemble docking strategy. It is possible to note that this strategy allowed us to select the correct binding pose (RMSD ≤ 2.0 Å) of all ligands with the lowest docking score (Table 5). Interestingly, despite donepezil being the smallest reference compound, the experimental binding mode was only found against its native AChE conformation (i.e., 4EY7). This can be due to the complete distinct chemical profile from the other tacrine-based ligands, inducing the enzyme to adopt a different conformation. Thus, the ensemble docking results reinforce the importance of considering multiple representative structures of AChE to consider the receptor flexibility, increasing the chance of finding the experimentally observed binding mode of ligands from different chemical classes.

## Figures and Tables

**Figure 1 molecules-25-03165-f001:**
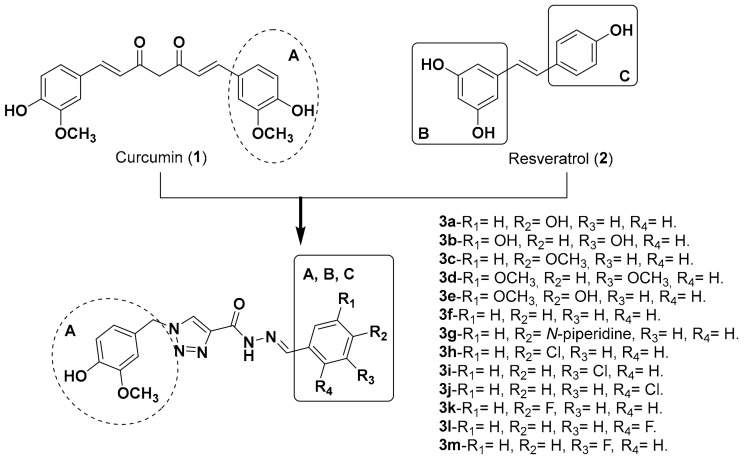
Planning by molecular hybridization of the molecular family synthesized in this work.

**Figure 2 molecules-25-03165-f002:**
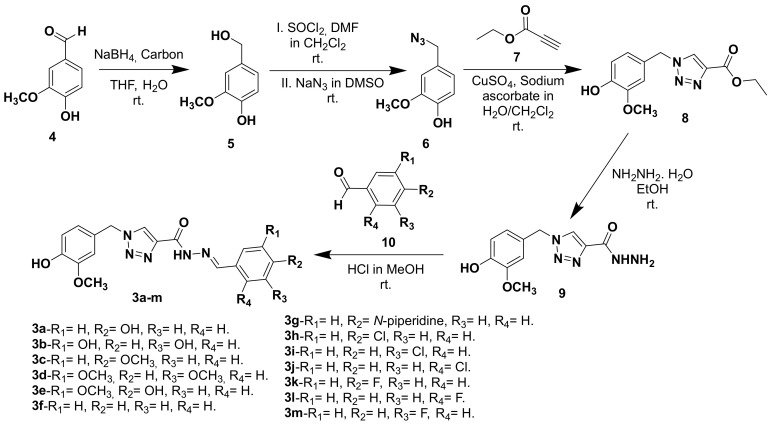
Synthetic route for the preparation of the target-compounds **3a**–**m.**

**Figure 3 molecules-25-03165-f003:**
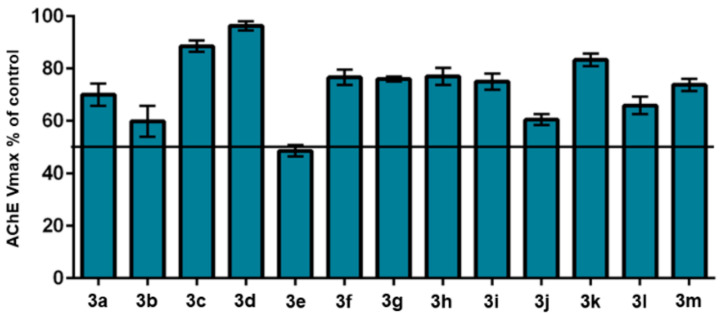
Effect of hybrid compounds on AChE activity (screened at 30 μM). The result is expressed as AChE maximum velocity (Vmax) in the presence of the indicated compound relative to uninhibited enzyme (control).

**Figure 4 molecules-25-03165-f004:**
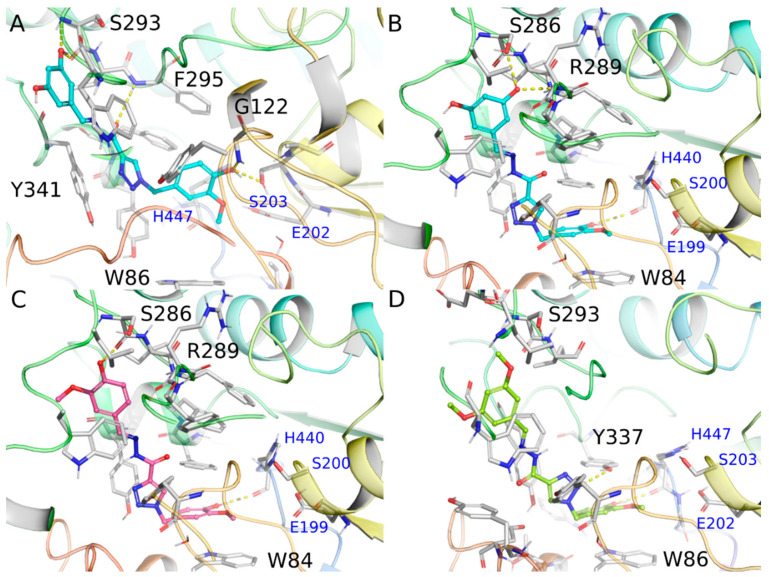
Docking results for compounds **3b** complexed with *4EY7* (**A**) and *1ZGC* (**B**), **3e** complexed with *1ZGC* (**C**), and **3d** complexed with *1Q84* (**D**). Hydrogen bonds are represented as yellow dashes.

**Figure 5 molecules-25-03165-f005:**
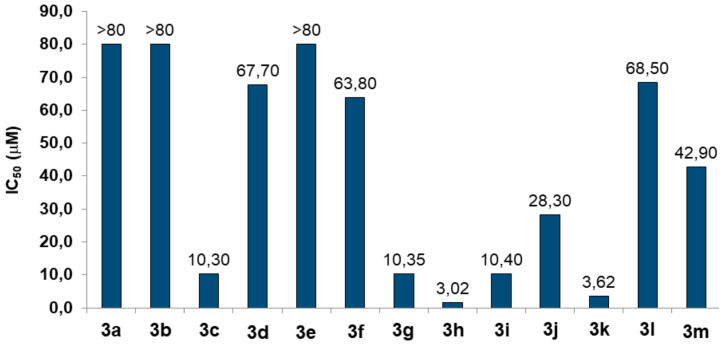
Toxicity of compounds **3a**–**3m** in neuronal SH-SY5Y cells. Cells were incubated for 24 h with different concentrations of the compounds (2.5–80 μM). At the end of incubation, the neuronal viability was measured using MTT assay. Data are reported as IC_50_ values (concentration resulting in 50% inhibition of neuronal viability) of the compounds.

**Figure 6 molecules-25-03165-f006:**
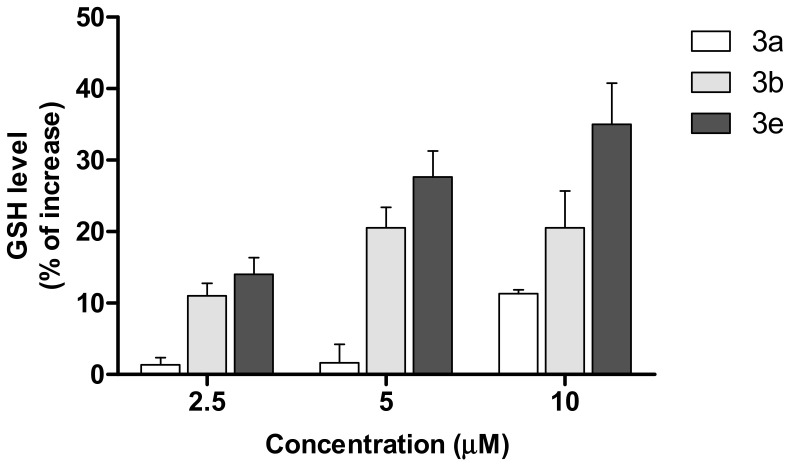
Effects of compounds **3a**, **3b** and **3e** on GSH levels in SH-SY5Y cells. Cells were incubated for 24 h with different concentrations of the compounds (2.5–10 μM). At the end of incubation, the GSH levels were measured using MCB assay. Data are expressed as a percentage of increase.

**Figure 7 molecules-25-03165-f007:**
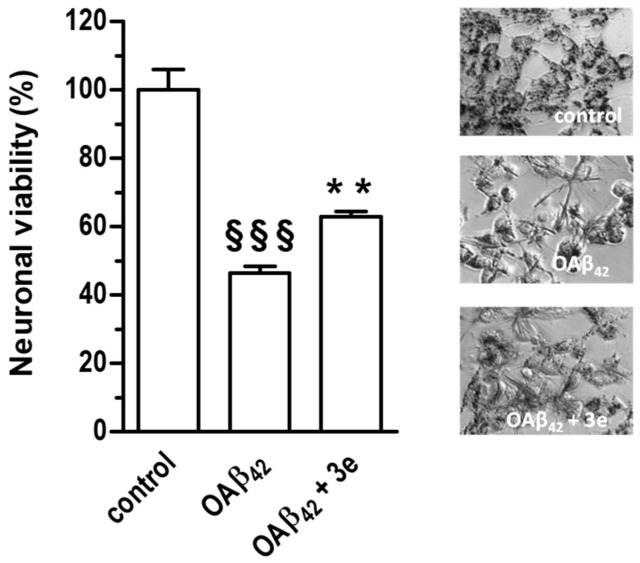
Compound **3e** counteracts the neurotoxicity induced by OAβ_42_ in SH-SY5Y cells. Cells were treated for 4 h with **3e** (5 μM) and OβA_42_ (10 μM). At the end of incubation, the neuronal viability was measured using MTT assay (^§§§^
*p* < 0.001 vs cells untreated; ** *p* < 0.01 vs cells treated with OAβ_42_ at one-way ANOVA with Bonferroni post hoc test). On the right representative images of formazan crystals from MTT in living SH-SY5Y cells.

**Figure 8 molecules-25-03165-f008:**
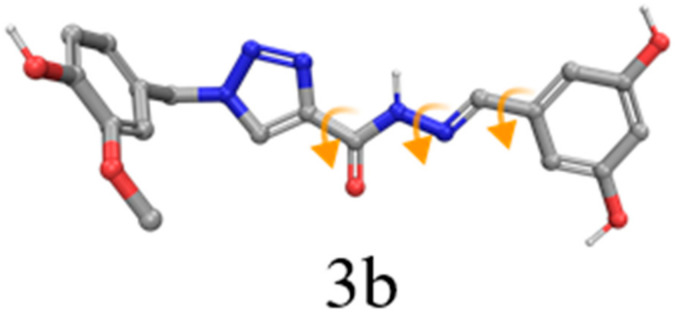
Compounds with rotatable bonds fixed during the docking experiments (orange arrows). Dihedrals not highlighted in this picture were defined as free to rotate, except amide bonds.

**Table 1 molecules-25-03165-t001:** Ensemble docking results of compounds **3b, 3e, 3d** and donepezil. The top-scored result of each compound is given for the AChE conformation that provided the best docking score according to the Glide XP scoring function (given in kcal/mol).

Compounds	*Ensemble Docking*				*Top-Scored*	
	*1ZGC*	*1Q84*	*2CKM*	*4EY7*	Score	PDB ID
**3b**	−10.67	−9.63	−8.19	−10.69	**−10.69/−10.67**	4EY7/1ZGC
**3d**	−8.79	−9.15	−8.47	−8.83	**−9.15**	1Q84
**3e**	−10.20	−9.39	−8.10	−8.78	**−10.20**	1ZGC
Donepezil	−11.14	−10.06	−12.57	−17.94	**−17.94**	4EY7

**Table 2 molecules-25-03165-t002:** ADME parameters predicted by computational studies.

Compounds	QPlogP o/w	HBA	HBD	PSA	% HOA	QPlogS	QPPCaco	QPlogBB
**3a**	1.73	7.3	3	124.8	72.28	-3.595	92.7	−2.09
**3b**	1.07	8.0	4	147.3	61.12	-3.282	36.2	−2.57
**3c**	2.48	7.3	2	111.2	85.85	-3.911	303.7	−1.56
**3d**	2.54	8.0	2	120.1	86.24	-4.056	303.7	−1.65
**3e**	1.79	8.0	3	133.0	73.14	-3.708	98.7	−2.15
**3f**	2.45	6.5	2	102.4	85.70	-3.817	303.2	−1.48
**3g**	3.58	7.5	2	104.4	92.35	-5.709	303.2	−1.63
**3h**	2.92	6.5	2	102.4	88.48	-4.519	303.2	−1.34
**3i**	2.84	6.5	2	102.4	87.96	-4.287	303.2	−1.34
**3j**	2.68	6.5	2	100.7	88.97	-3.839	388.8	−1.30
**3k**	2.67	6.5	2	102.4	86.99	-4.154	303.2	−1.38
**3l**	2.47	6.5	2	102.1	85.90	-3.837	305.7	−1.46
**3m**	2.62	6.5	2	102.4	86.73	-4.045	303.2	−1.39

**QPlogP o/w**: Predicted octanol/water partition coefficient (−2.0 to 6.5). **HBA:** Hydrogen bonding acceptors (2 to 20). **HBD**: Hydrogen bonding donor (0 to 6). **PSA**: Van der Waals surface area of polar nitrogen and oxygen atoms (7 to 200). **% HOA**: Percentage of human absorption by oral route (<25%-low; >80%-high). **QPlogS**: Aqueous solubility (−6.5 to 0.5); **QPPCaco:** Permeability in Caco-2 cell assay, model for intestinal absorption (<25–low; >500–high); **QPlogBB**: Permeability in the blood-brain barrier (−3.0 to 1.2).

**Table 3 molecules-25-03165-t003:** Antioxidant activity of compounds against the DPPH radical and ROS formation induced by *t*-BOOH in neuronal SH-SY5Y cells ^a^.

Compound	DPPH EC_50_ (mg/L)	Direct Antioxidant Activity in SH-SY5Y Cells ^b^	Indirect Antioxidant Activity in SH-SY5Y Cells ^c^
**3a**	>200	In ^d^	20.58
**3b**	>200	In	14.07
**3c**	>200	-	-
**3d**	>200	-	-
**3e**	30.44	29.79	35.16
**3f**	>200	-	-
**3g**	81.51	-	-
**3h**	131.5	-	-
**3i**	190.9	-	-
**3j**	>200	-	-
**3k**	59.00	-	-
**3l**	138.5	-	-
**3m**	62.65	-	-
Ascorbic acid	14.92	-	-
Trolox	27.76	-	-

^a^ The antioxidant activity is expressed as percentage inhibition of ROS formation induced by *t*-BOOH in SH-SY5Y cells; ^b^ the direct antioxidant activity was evaluated after a simultaneously treatment (30 min) with the compound (10 µM) and *t*-BOOH (100 μM); ^c^ the indirect antioxidant activity was evaluated after a chronic treatment (24 h) with the compound (10 μM) then exposed to *t*-BOOH (100 μM for 30 min); ^d^ In: inactive.

**Table 4 molecules-25-03165-t004:** Conserved waters considered in the docking experiments.

Water	1ZGC	1Q84	2CKM	4EY7
Wat1	1468	1708	2062	729
Wat2	1481	1755	2054	737
Wat3	1489	1715	2061	722
Wat4	1531	1735	2035	731

**Table 5 molecules-25-03165-t005:** Ensemble docking results for the reference ligands against the four AChE conformations. The binding poses selected according to the ensemble docking strategy (i.e., the AChE conformation that provides the lowest XP Score pose) are highlighted in bold, achieving 100% of success rate when considering RMSD ≤ 2.0 Å.

Reference Ligand	1ZGC		1Q84		2CKM		4EY7	
	Score	RMSD	Score	RMSD	Score	RMSD	Score	RMSD
1ZGC	−18.873	1.411	−18.553	1.515	**−18.989**	**1.618**	−17.273	12.382
1Q84	−18.246	1.439	**−19.085**	**0.833**	−12.208	4.773	−11.546	6.852
2CKM	−17.954	4.255	−16.969	3.608	**−20.525**	**1.234**	−13.255	5.905
4EY7	−11.136	2.436	−10.064	3.250	−12.575	4.235	**−17.941**	**0.594**
Success Rate	50%		50%		50%		25%	

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
