# Peer review of "Design, Synthesis and Biological Evaluation of Novel Triazole N-acylhydrazone Hybrids for Alzheimer’s Disease"

_molecules, 2020, doi:10.3390/molecules25143165_

Round 1

Reviewer 1 Report

The manuscript overall can fit the aims and scope of the special issue since it describes the synthesis and biological evaluation of N-acyl-hydrazone function linked to a 1,4-disubstituted triazole system which include structural subunits derived from curcumin and resveratrol . Among these hybrid compounds, derivative 3e was shown to inhibit the acetylcholinesterase activity, the intracellular formation of reactive oxygen species as well as the neurotoxicity elicited by Aβ42 oligomers in neuronal SHSY5Y cells. In parallel, compound 3e showed a good profile of safety and ADME parameters.

Overall the manuscript is nicely written although there are some problems with the spaces throughout the whole text (see next to some[ref] and next to IC50= ......). Moreover, some grammar editing is needed i.e. "Compound 3e was identified as the most active in the series, inhibiting more than 50% of the AChE activity at the evaluated concentration and exhibiting an IC50 value of 20.22 μM, showing shown to be better than Curcumin" or "the direct antioxidant activity of all compounds 3a-m (1.56-200 mg/L) were was assessed against the DPPH radical.." among others.

Please use the abbreviation Aβ42 for the amyloid peptide as it is conventionally written.

The synthesis part is good and a reasonable number of molecules is described. PLease rephrase the synthesis section omitting the : and the numbering of the steps a)....b)....c)..... Furthermore, add the appropriate references in this section as they appear in the experimental section. Figure 1, please simplify the hybridisation picture (the two squares and the oval shapes which overlap in the right hand side of the molecule are too much too confusing)

It is interesting that the diphenyl derivative 3b resulted in smaller direct and indirect antioxidant effect. The author should add a small discussion on this.

Figure 7. - please remove B since it does not add any information and it is not mentioned in the text.

The legend of Figure 7 reports " Cells were treated for 4 h with 3e [5 μM] and OβA1-42 [10 μM]. At the end of incubation, the neuronal viability was measured using MTT assay" and the experimental section mentions "SH-SY5Y cells were seeded in a 96-well plate at 3 × 104 cells/well, incubated for 24 h, and treated with the compounds for 24 h. Then, cells were treated with Aβ1-42 oligomers for 4 h. The neuroprotective activity, in terms of increase in intracellular MTT granules, was measured by MTT assay, as previously described". Which one is the correct experimental protocol??? If a preincubation of the compound with the cells occur then the effect observed by the MTT may only be the result of the antioxidant activity. More experiments, such as CD, thioflavin T fluorescence, TEM etc are needed to support the notion "The results suggest that hybrid compound 3e can halt the interaction between the OßA1-42 and the neuronal membrane, an early step of the neurotoxicity mediated by OßA1-42. Interestingly, this hybrid compound preserve the ability of curcumin [29] and resveratrol [21] functional groups to inhibit the aggregation of βA." If this investigation is not performed then the authors can not claim :"...as well as the neurotoxicity elicited by Aβ42 oligomers in neuronal SHSY5Y cells." especially when the rescuing effect is statistically significant but still very small (only about 15%) and there is plenty of reports for molecules which inhibit Abeta aggregation and result in rescuing neuronal cells form Abeta-induced toxicity.

"Considering that the neuroinflammation is a pathogenic event in AD linked to OS, compound 3e was further assayed on inflammation elicited by lipopolysaccharide (LPS) in human monocyte THP-1 cells differentiated to microglial cells with phorbol 12-myristate 13-acetate (PMA) [30]. In particular, the gene expression of several inflammation markers were evaluated, such as iNOS, IL-1β, TNF-α and COX-2 [31]. Compound 3e did not show any anti-inflammatory activity (data not shown)." There is a full experimental description and no data at all. Please add some data to the supporting information or remove completely. The same for VERO cell cytotoxicity. 

Author Response

Response to Reviewer 1 Comments

We thank the reviewer for the critical appraisal and suggestions and we modified the text in accordance with scientific statements.

The manuscript overall can fit the aims and scope of the special issue since it describes the synthesis and biological evaluation of N-acyl-hydrazone function linked to a 1,4-disubstituted triazole system which include structural subunits derived from curcumin and resveratrol . Among these hybrid compounds, derivative 3e was shown to inhibit the acetylcholinesterase activity, the intracellular formation of reactive oxygen species as well as the neurotoxicity elicited by Aβ42 oligomers in neuronal SHSY5Y cells. In parallel, compound 3e showed a good profile of safety and ADME parameters.

Overall the manuscript is nicely written although there are some problems with the spaces throughout the whole text (see next to some[ref] and next to IC50= ......). Moreover, some grammar editing is needed i.e. "Compound 3e was identified as the most active in the series, inhibiting more than 50% of the AChE activity at the evaluated concentration and exhibiting an IC50 value of 20.22 μM, showing shown to be better than Curcumin" or "the direct antioxidant activity of all compounds 3a-m (1.56-200 mg/L) were was assessed against the DPPH radical.." among others.

ANSWER: we corrected the mistakes and rephrased this sentence.

Please use the abbreviation Aβ42 for the amyloid peptide as it is conventionally written.

ANSWER: we thank the reviewer for the suggestion. We used the abbreviation Aβ42 in both the text and the Figure 7.

The synthesis part is good and a reasonable number of molecules is described. Please rephrase the synthesis section omitting the: and the numbering of the steps a)....b)....c).... Furthermore, add the appropriate references in this section as they appear in the experimental section. Figure 1, please simplify the hybridisation picture (the two squares and the oval shapes which overlap in the right hand side of the molecule are too much too confusing)

ANSWER: we rephrased the various steps of synthesis and added the appropriate references. Further, we improved the hybridisation picture (Figure 1).

It is interesting that the diphenyl derivative 3b resulted in smaller direct and indirect antioxidant effect. The author should add a small discussion on this.

ANSWER: we added a small discussion on hybrid compounds, including 3b in section 2.2.4.

Figure 7. - please remove B since it does not add any information and it is not mentioned in the text.

ANSWER: we removed B from Figure 7 and its legend.

The legend of Figure 7 reports " Cells were treated for 4 h with 3e [5 μM] and OβA1-42 [10 μM]. At the end of incubation, the neuronal viability was measured using MTT assay" and the experimental section mentions "SH-SY5Y cells were seeded in a 96-well plate at 3 × 10 cells/well, incubated for 24 h, and treated with the compounds for 24 h. Then, cells were treated with Aβ1-42 oligomers for 4 h. The neuroprotective activity, in terms of increase in intracellular MTT granules, was measured by MTT assay, as previously described". Which one is the correct experimental protocol??? If a preincubation of the compound with the cells occur then the effect observed by the MTT may only be the result of the antioxidant activity.

ANSWER: we treated the cells for 4 h with 3e and OAβ42. We therefore corrected the mistake in experimental section.

More experiments, such as CD, thioflavin T fluorescence, TEM etc are needed to support the notion "The results suggest that hybrid compound 3e can halt the interaction between the OßA1-42 and the neuronal membrane, an early step of the neurotoxicity mediated by OßA1-42. Interestingly, this hybrid compound preserve the ability of curcumin [29] and resveratrol [21] functional groups to inhibit the aggregation of βA." If this investigation is not performed then the authors can not claim :"...as well as the neurotoxicity elicited by Aβ42 oligomers in neuronal SHSY5Y cells." especially when the rescuing effect is statistically significant but still very small (only about 15%) and there is plenty of reports for molecules which inhibit Abeta aggregation and result in rescuing neuronal cells form Abeta-induced toxicity.

ANSWER: we thank the reviewer for these critical comments. We deleted the discussion that was not supported by data in section 2.2.5. We further removed the sentence “.. as well as the neurotoxicity elicited by Aβ42 oligomers in neuronal SHSY5Y cells” and rephrased the sentence focusing on the antioxidant mechanisms in conclusion section.

"Considering that the neuroinflammation is a pathogenic event in AD linked to OS, compound 3e was further assayed on inflammation elicited by lipopolysaccharide (LPS) in human monocyte THP-1 cells differentiated to microglial cells with phorbol 12-myristate 13-acetate (PMA) [30]. In particular, the gene expression of several inflammation markers were evaluated, such as iNOS, IL-1β, TNF-α and COX-2 [31]. Compound 3e did not show any anti-inflammatory activity (data not shown)." There is a full experimental description and no data at all. Please add some data to the supporting information or remove completely. The same for VERO cell cytotoxicity.

ANSWER: we thank the reviewer for these comments. The negative results recorded with neuroinflammation assay and VERO cell cytotoxicity do not add value to manuscript. We therefore deleted these informations.

Reviewer 2 Report

The Authors designed and synthesized a group of novel triazole-based compounds as possible anti-Alzheimer disease agents. The obtained compounds possessed quite weak activity, however they exerted multifunctional properties. In my opinion the Authors should revise their paper before publication. The following concerns should be addressed:

1) possible toxicity against VERO cells should be described or it should not be mentioned in the manuscript (subchapter 4.2.5 should be deleted)

2) determination of the anti-inflammatory activity measured by a quantitative real-time PCR (described in the experimental subchapter 4.2.9) would be valuable element of the manuscript. Since the Authors did not mention the obtained results, therefore subchapter 4.2.9 has no sense at all.

3) I recommend the Authors change the term "neurotoxicity" (subchapter 2.2.2) into cytotoxicity/cellular toxicity or simply toxicity against human neuronal (SH-SY5Y) cells (because neurotoxicity is much more complex phenomenon than the ability of the compound to inhibit the growth of the respective cells).

Author Response

Response to Reviewer 2 Comments

We thank the reviewer for the critical appraisal and suggestions and we modified the text in accordance with scientific statements.

The Authors designed and synthesized a group of novel triazole-based compounds as possible anti-Alzheimer disease agents. The obtained compounds possessed quite weak activity, however they exerted multifunctional properties. In my opinion the Authors should revise their paper before publication. The following concerns should be addressed:

  • possible toxicity against VERO cells should be described or it should not be mentioned in the manuscript (subchapter 4.2.5 should be deleted)

ANSWER: we thank the reviewer for this comment. The VERO cell cytotoxicity does not add value to manuscript. We therefore deleted this information.

  • determination of the anti-inflammatory activity measured by a quantitative real-time PCR (described in the experimental subchapter 4.2.9) would be valuable element of the manuscript. Since the Authors did not mention the obtained results, therefore subchapter 4.2.9 has no sense at all.

ANSWER: we thank the reviewer for this comment. The negative results recorded with neuroinflammation assay do not add value to manuscript. We therefore deleted these informations.

  • I recommend the Authors change the term "neurotoxicity" (subchapter 2.2.2) into cytotoxicity/cellular toxicity or simply toxicity against human neuronal (SH-SY5Y) cells (because neurotoxicity is much more complex phenomenon than the ability of the compound to inhibit the growth of the respective cells).

ANSWER: we thank the reviewer for the suggestion. We changed the term “neurotoxicity” into “toxicity”.

Reviewer 3 Report

Dear authors,

This article describes the chemical synthesis and biological activity of a variety of triazole N-acylhydrazone hybrids. The synthesis is well established and the synthesized compounds are well characterized.

Regarding the biological activity, the results seems interesting. However, I missed along the manuscript more explanations related to the chemical structure and the bioactivity. I consider that this is the main aspect of the work and thus I suggest to write a sentence in the “Conclusion” part indicating which structural characteristic are more appropriate for the biological activity studied.

Apart from this, please check the following items:

TABLE 1

Score -10.68/-10.67 or -10.69/-10.67?

Line 129

-10.68 or -10.69 Kcal/mol?

TABLE 2

AHB or HBA? and DHB or HBD?. Please, unify

Experimental

HR-MS (ESI) m/z: Calcd for [M + H]+ 382.1515 g/mol, found 382.1494 g/mol.

Is this the Molecules format for HR-MS?. I do not think so. Please, check this point.

Author Response

Response to Reviewer 3 Comments

We thank the reviewer for the critical appraisal and suggestions and we modified the text in accordance with scientific statements.

Dear authors,

This article describes the chemical synthesis and biological activity of a variety of triazole Nacylhydrazone hybrids. The synthesis is well established and the synthesized compounds are

well characterized. Regarding the biological activity, the results seems interesting. However, I missed along the manuscript more explanations related to the chemical structure and the bioactivity. I consider that this is the main aspect of the work and thus I suggest to write a sentence in the “Conclusion” part indicating which structural characteristic are more appropriate for the biological activity studied.

ANSWER: we thank the reviewer for this suggestion. We added more informations about the relationship between the chemical structures of the hybrid compounds and their bioactivity in section 2.2.4. as well as in conclusion section.

Apart from this, please check the following items:

TABLE 1 Score -10.68/-10.67 or -10.69/-10.67?

Line 129 -10.68 or -10.69 Kcal/mol?

TABLE 2 AHB or HBA? and DHB or HBD?. Please, unify

ANSWER: we corrected these mistakes.

HR-MS (ESI) m/z: Calcd for [M + H]+ 382.1515 g/mol, found 382.1494 g/mol.

Is this the Molecules format for HR-MS?. I do not think so. Please, check this

point.

ANSWER: Molecules journal does not provide a specific format for mass spectrometry data. We therefore left our format.

Round 2

Reviewer 1 Report

Still some minor editing is needed e.g. IC50 = 24 mM

Author Response

We thank the reviewer for detecting the mistake. In this regard we did not find the highlighted mistake IC50 = 24 mM along the text. The reviewer may provide additional information to detect this mistake.

Reviewer 2 Report

I accept the changes made by the Authors in the manuscript. Now it is suitable for publication.

Author Response

Thanks